# Awareness and Perceptions of the Impact of Tonsillectomy on the Level of Immunity and Autoimmune Diseases among the Adult Population in Abha City, Kingdom of Saudi Arabia

**DOI:** 10.3390/healthcare11060890

**Published:** 2023-03-20

**Authors:** Ayoub A. Al-shaikh, Abdullah Alhelali, Syed Esam Mahmood, Fatima Riaz, Abdulrahim Ali Hassan Hassan, Abduaelah Ali H Hassan, Bandar Mohammed Mushabbab Asiri, Abdulaziz Saad Mohammed Al-shahrani, Abdullah Jallwi Mohammed Korkoman, Abdullah Fahad Alahmari, Abeer Ali Hassan Hassan, Mohammed O. Shami, Ausaf Ahmad, Rishi K. Bharti, Md. Zeyaullah

**Affiliations:** 1Department of Family and Community Medicine, College of Medicine, King Khalid University, Abha 62529, Saudi Arabia; 2Department of Otolaryngology, Head and Neck Surgery, Aseer Central Hospital, Abha Children Hospital, Abha 62523, Saudi Arabia; 3College of Medicine, King Khalid University, Abha 62529, Saudi Arabia; 4College of Medicine, Jazan University, Jazan 45142, Saudi Arabia; 5Department of Community Medicine, Integral Institute of Medical Science and Research, Integral University, Kursi Road, Lucknow 226026, India; 6Department of Basic Medical Science, College of Applied Medical Sciences, Khamis Mushayt Campus, King Khalid University (KKU), Abha 62561, Saudi Arabia

**Keywords:** awareness, perceptions, tonsillectomy, immunity

## Abstract

The widespread misconception that tonsillectomy leads to a decrease in immunity may lead to fear and avoidance of the operation. This can result in a deterioration of the situation, such as sleep-related breathing issues, frequent infections, and an increase in complications. The current research was conducted to assess the awareness and perception with respect to the impact of tonsillectomy on the immune system and to assess the awareness and perception of the relationship between autoimmune diseases and tonsillectomy. This 6-month descriptive cross-sectional online questionnaire survey was conducted among individuals who were 18 years and above living in Abha city, Saudi Arabia. Out of the 800 study subjects, 104 (13%) had undergone tonsillectomy. Statistically significant associations were found between age group, education, income, and occupation among those who had undergone tonsillectomy. Multivariate logistic regression analysis showed that ages 18–30 years and 31–40 years (OR: 2.36, 95% CI: 1.18–4.71, and OR: 1.46, 95% CI: 0.53–3.97) and education levels of high school, bachelors, and above (OR: 8.30, 95% CI: 3.05–22.58 and OR: 10.89, 95% CI: 4.23–28.05) were found to be associated with tonsillectomy status of the subjects. On the contrary, income levels of 5000–9000 and >9000 (OR: 0.65, 95% CI: 0.36–1.17 and OR: 0.78, 95%CI: 0.42–1.42) and male gender (OR: 0.79, 95% CI: 0.52–1.19) were found to be associated with non-tonsillectomy status of subjects. Almost 36% of study subjects thought that tonsillectomy affects immunity. Only 18% of study subjects thought that there is a relationship between tonsillectomy and autoimmune diseases. About one-third of the respondents had received this information from community members and social media. A small number of study subjects relied on public awareness programs. Therefore, social media can play a vital role in the community to remove misconceptions regarding tonsillectomy and its effect on immunity and autoimmune disease. Further educational interventional studies are required to see the correction to the public perception of tonsillectomy and its effect on immunity and autoimmune diseases.

## 1. Introduction

The surgical field is growing in this scientific era, and tremendous advancements have been achieved related to surgeries and techniques. Otorhinolaryngology is also one of the surgical fields with a lot of advancements. A lot of procedures are being conducted in Ear, Nose, and Throat (ENT) fields, and tonsillectomy is one of the most common surgeries conducted in this field. Tonsillectomy is a surgical procedure executed either with adenoidectomy or without adenoidectomy, with complete removal of the tonsillar tissue [1]. The palatine tonsils are lympho-epithelial biological structures found at the junction of the mouth and oropharynx. Recent literature also suggests that tonsils play an important role in the body’s immune system functioning. The tremendous immunologic function of the tonsils is initiated between the ages of 3 and 10 years. As a result, the tonsils’ function is quite important between 3 and 10 years of age, and they also show age-dependent involution thereafter as well [2]. It remains disputed whether tonsillectomy decreases immunity levels [3], while some studies suggest that there is no impact of the removal of tonsils on the immune system and immune functions [4]. Some research has found evidence of some short-term impacts on the removal of tonsils on the immune system [5,6]. However, significant evidence is not available suggestive of the long-term effects of tonsillectomy on the immune system [7,8]. A total of 179,875 individuals in Sweden underwent tonsillectomy, of whom 5357 subsequently developed autoimmune diseases. Some research also suggests that tonsillectomy leads to the development of autoimmune diseases later in life, and there is literature available showing a scientific association of tonsillectomy with the development of autoimmune diseases. Studies have reported that the incidence of diseases related to autoimmunity was more prevalent among individuals who had undergone a tonsillectomy. Immune dysfunction because of tonsillectomy may partly explain the observed association. However, the occurrences of autoimmune diseases were inflated among individuals who had undergone a tonsillectomy, as proven by performing certain laboratory tests such as chemistry and immunologic analysis. Tonsils play an important role in immunity and defense against bodily infections and foreign pathogens. When antigens are inhaled or ingested, tonsils are positioned for exposure, which can lead to the development of lymphokines and immunoglobulins. Composed predominately of B-cell lymphoid tissue, one of the roles served by tonsils is that of mucosal secretory immunity. On the surface of the tonsils, one can find specialized antigen-capture cells referred to as M cells. These cells permit the capture of antigens generated by microorganisms. After recognizing an antigen, the M cells activate T and B cells in the tonsils and trigger an immune response. B cells, when stimulated, proliferate in the germinal areas of the tonsils. At the germinal center, B memory cells mature and are stored for repeated exposure to the same antigen. B cells also serve to secrete IgA, an antibody that plays a vital role in the immune function of mucus [9]. The basic mechanisms of the development of autoimmune diseases need to be explored in upcoming studies [10]. Overall, there are many controversies related to the post-tonsillectomy effects on immunity and autoimmune diseases. Being the most common ENT procedure, its prevalence, cost, and impact on the body suggest a great need for evidence-based guidelines, policies, and their implementation [11]. Previous studies have reported fluctuations in the rate of adenotonsillectomy because of the lack of implementation policies. In some places, tonsillectomies are over-performed, and in other places, they are under-performed [12]. To date, the postoperative effects of tonsillectomies on the immune response have been debated among physicians and are of great concern for parents of children undergoing tonsillectomy [4]. There is also limited evidence of providing health education and public awareness on this topic. Public awareness about tonsillectomy also plays a crucial role in the frequency of the procedure and parental satisfaction. Poor public awareness about tonsil and adenoid hypertrophy is also reported by head and neck surgeons [13].

The available data on public awareness regarding the effect of tonsillectomy on immunity and the development of autoimmune diseases, including factors that influence the decision of parents about this surgery, is very scarce [10]. There is a widespread misconception among the community that has been observed during clinical practice, which is that tonsillectomy leads to a decrease in immunity, which may lead to fear and avoidance of the operation, leading to a deterioration of the situation and an increase in complications as well. Therefore, further research is necessary to assess community-level awareness about the relationship between tonsillectomy and the level of immunity and development of autoimmune diseases in the Kingdom of Saudi Arabia. Because of the unavailability of such research in Saudi Arabia and especially in Abha city, along with this background, this study was conducted to assess the awareness and perception regarding the effect of tonsillectomy on the immune system and to assess the awareness and perception of the relationship between autoimmune diseases and tonsillectomy among adults in Abha city of Saudi Arabia. We hypothesize that there is a low level of awareness and perception in the adult population regarding the impact of tonsillectomy on the level of immunity and autoimmune diseases.

## 2. Material and Methods

A descriptive cross-sectional online questionnaire survey was conducted in Abha, a city in the Aseer region of Saudi Arabia. The general population of 18 years of age and above, who were residing in the study area during the period of study, i.e., May 2022 to December 2022, were included. The people who refused to participate were excluded.

### 2.1. Sample Size

The estimated sample size of the study population was calculated using a 50% response distribution, a 3.4% margin of error, and a 95% confidence interval. The estimated sample size was calculated as 800 participants [14,15].

Based on the literature review, the researchers constructed a questionnaire for the purpose of the study to avoid errors in data collection. The questionnaire was further reviewed by subject experts and was disseminated online after developing thorough Google forms for the participants. The questionnaire consisted of close-ended questions constructed both in English and Arabic language. The questionnaire was translated from English language to Arabic language (which is the local language) by a person who was bilingual. This enabled an easy understanding of the study question by local participants of the study and avoided bias.

The questionnaire was distributed through social media and e-mail for the convenience of data collection and to avoid face-to-face interviews which were more time-consuming than the online survey questionnaire.

A pilot study of 25 individuals was conducted to assess the questionnaire’s validity, reliability, applicability, and average filling time before the administration of the final version of the questionnaire. The reliability coefficient (α-Cronbach’s) was 0.76. The questionnaire contained 17 items. It was divided broadly into 3 sections: 1—demographic information, 2—medical history of tonsillectomy, and 3—awareness regarding tonsillectomy and autoimmune diseases.

### 2.2. Sampling Technique

The participants were selected by convenience through a call for answer to a wide audience, with subjects answering based on their own thoughts.

### 2.3. Informed Consent

The questionnaire started with a brief explanation of the objectives of the study and intended to remind the participants that their participation in the study was entirely their own choice. Names of participants were not collected, and their identity was kept confidential and anonymous. An electronic version of an informed consent form was obtained from all study subjects.

### 2.4. Ethical Considerations

Ethical approval (ECM#2022-2803) was obtained from the Research Ethics Committee of King Khalid University on 12-10-2022. Participants were assured that their data would be kept anonymous, confidential, and utilized only for research purposes. The data were kept in a password-protected cloud system for safety purposes. The use of anonymous data in this research project was reviewed and approved by the research ethics committee.

### 2.5. Statistical Analysis

The collected data were coded and then entered into an Excel sheet (Microsoft Office Excel 2010) database. The data were analyzed using SPSS (Statistical Package for Social Sciences), version 16.0 (SPSS, Inc., Chicago, IL, USA). The descriptive variables were presented using frequency, percentage, and graphs as appropriate. Pearson’s chi-square test was used to assess the association between participants who had tonsillectomy as a dependent variable and related risk factors as independent variables. Binary logistic regression models were fit to calculate odds ratio. Multicollinearity was tested by a variance inflation factor calculation. A *p*-value of less than 0.05 was regarded as statistically significant.

## 3. Results

Table 1 shows the association between sociodemographic factors and subjects who had undergone tonsillectomy. Out of the 800 study subjects, 104 (13%) had undergone a tonsillectomy, and 696 (87%) had never had a tonsillectomy. The ratio of male to female study subjects was 9:11. Almost 86% of study subjects belonged to the age group of 18 to 30 years. About two-thirds of the study subjects were students. Statistically significant associations were found between age group, education level, income and occupation, and those who had undergone tonsillectomy.

Table 2 illustrates the association of knowledge about the relationship between tonsillectomy and autoimmune diseases among subjects who had a tonsillectomy. Queries such as ‘Do you think tonsillectomy affects immunity?’ showed statistically significant associations among subjects who had undergone a tonsillectomy. Almost 13% of study subjects think that tonsillectomy affects immunity. Only 15.3% of study subjects think that there is a relationship between tonsillectomy and autoimmune diseases.

Multivariate logistic regression analysis showed that ages 18–30 years and 31–40 years (OR: 2.36, 95% CI: 1.18–4.71, and OR: 1.46, 95% CI: 0.53–3.97) and education levels of high school and Bachelors and above (OR: 8.30, 95% CI: 3.05–22.58 and OR: 10.89, 95% CI: 4.23–28.05) were found to be associated with the tonsillectomy status of subjects.

On the contrary, income level, 5000 to 9000 and more than 9000 (OR: 0.65, 95% CI: 0.36–1.17 and OR: 0.78, 95%CI: 0.42–1.42), and male gender (OR: 0.79, 95% CI: 0.52–1.19), were found to be associated with a non-tonsillectomy status of subjects (Table 3). 

Figure 1 shows the frequency distribution of the source of information related to the query ‘Do you think tonsillectomy affects immunity’. Community members (286, 35.8%) and social media (261, 32.6%) were the major sources of information, followed by health practitioners (135, 16.9%). A very small number of study subjects relied on traditional media (39, 4.9%).

Figure 2 shows the percent distribution of sources of information related to the query ‘is there a relationship between tonsillectomy and autoimmune diseases’. Social media (33.8%) and community members (31.8%) were the major sources of information, followed by a health practitioner (17.1%). A small number of study subjects received information from traditional media (2.9%).

Figure 3 shows the distribution of perceptions of the study subjects regarding the type of infection in the long term that can affect a person after a tonsillectomy has been performed. Among all the types of infections, the majority of the study subjects responded to respiratory infection (70.5%), followed by gastrointestinal infection (13.8%) and cardiovascular infection (7.9%).

## 4. Discussion

Tonsillectomy is one of the common surgeries performed by ENT surgeons for certain indications. Researchers suggest that tonsils play a great role in the evolution of immunity, hence why tonsillectomy affects our immune system. It is also suggested by researchers that a tonsillectomy increases the risk of the development of autoimmune diseases as well [9]. Tonsillar tissues act as a first-line defense mechanism of the immune system against organisms, allergens, and even food [16].

In this study, we explored public views about the relationship of tonsillectomy with the effect on the immune system and the development of autoimmune diseases. A previous study, conducted only on tonsillectomy patients, had a mean age of about 16 years, which is similar to our study findings that show about 23.5% of our samples with tonsillectomy were less than 18 years of age [15]. Tonsillectomy was found to be common among school-going children who were mostly in middle school (57.9%), whereas a study conducted in Australia, by Tran AH et al., showed a high prevalence of tonsillectomy among children 5–9 years of age [17]. Around 39.2% of patients with tonsillectomy have a family income of up to 9000 SAR, suggesting a lower middle class, which is also supported by research showing that twice as many children with age <16 years belong to the lower socioeconomic class [11,18,19]. However, household income showed a weak association with tonsillectomy in many other studies [20,21].

About 87% of our respondents who had never had tonsillectomy considered that tonsillectomy affects immunity, whereas 13% of respondents believed that tonsillectomy affects immunity, indicating that the rate is lower among those who had undergone tonsillectomy themselves. A study conducted by Tai KH et al. about the parent’s perspective on tonsillectomy on social media suggested that one of the views that parents held was that even after tonsillectomy, their child would get sick again [22]. However, literature indicated that tonsillectomy performed on patients with recurrenttonsillitis results in better quality of life and reduced incidence of upper respiratory tract infections and tonsillitis [23].

However, another study conducted in Iran does not suggest any difference in immunity levels among patients with tonsillectomy [24], whereas another study showed reduced humoral parameters after tonsillectomy, but the overall impact on humoral immunity was not significantly reduced [3]. Another study also supported that tonsillectomy does not impose a negative short-term or long-term effect on the cellular and humoral immunity among children [25].

A vast majority of our respondents believe that tonsillectomy is associated with autoimmune diseases. A study conducted in Sweden among tonsillectomy patients from the period of 1997–2012 showed an overall Standardized Incidence Ratio (SIR) of 1.34 (95% CI: 1.30 to 1.37), and that the incidence of autoimmune diseases was higher among individuals with tonsillectomy. However, basic causative mechanisms need to be explored further [9]. Tonsillectomy is associated with numerous autoimmune diseases such as alopecia aerate [26], periodontitis [27], irritable bowel disease [28], and multiple sclerosis [29]. However, tonsillectomy may serve some protective role in the development of IgA nephropathy [30,31] and homozygous HLA-Cw* 0602 genomic carriage in patients with plaque psoriasis, and may predict a good outcome post-tonsillectomy [32]. A population-based cohort study also suggested a decreased risk of the development of psoriasis among patients with tonsillectomy [33].

Around 13% of our respondents who had tonsillectomy performed before would not want to suggest tonsillectomy for someone else based on its relationship to immunity.

However, there is more susceptibility to sleep apnea before surgical removal of tonsils, which is also associated with increased scores on the post-consultation obstructive sleep disorders breathing and adenotonsillectomy knowledge score (OR, 4.07; 95% CI: 1.17 to 16.17) [34].

Our study showed that the most common source of information regarding tonsillectomy and immunity was community members (35.8%) and social media (32.6%), followed by a health practitioner (16.9%). A very small number of study subjects relied on traditional media (4.9%), whereas the most common source for information on the association of tonsillectomy and autoimmune diseases was social media (33.8%). These findings highlighted the importance of social media as a tool for community awareness on a mass level. Public awareness programs may bring a positive change in community behavior regarding health and disease. Public posts on social media regarding tonsillectomy give a chance for parents and the general population a way to express themselves and obtain other opinions as well. Sources of social media may help in enhancing patient-centered approaches and their impact on health care [35]. The study limitations are that since the recruitment was performed online, it could be a biased study; the cross-sectional nature of this study cannot confirm the causality association between the compared variables, and the self-reported responses could over or underestimate the results.

## 5. Conclusions

Public awareness programs and social media can play a vital role in the community to remove the misconceptions regarding tonsillectomy and its effect on immunity and autoimmune diseases. Further studies are also required with long-term follow-up to decide the level of the immune system after surgery. Moreover, further educational interventional studies are also required to see the removal of misconceptions in the public perception about tonsillectomy and its effect on immunity and autoimmune diseases.

## Figures and Tables

**Figure 1 healthcare-11-00890-f001:**
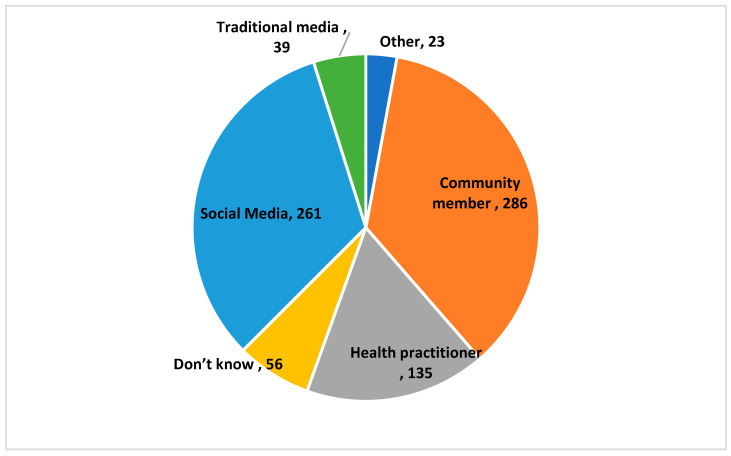
Source of information among respondents regarding tonsillectomy affecting immunity.

**Figure 2 healthcare-11-00890-f002:**
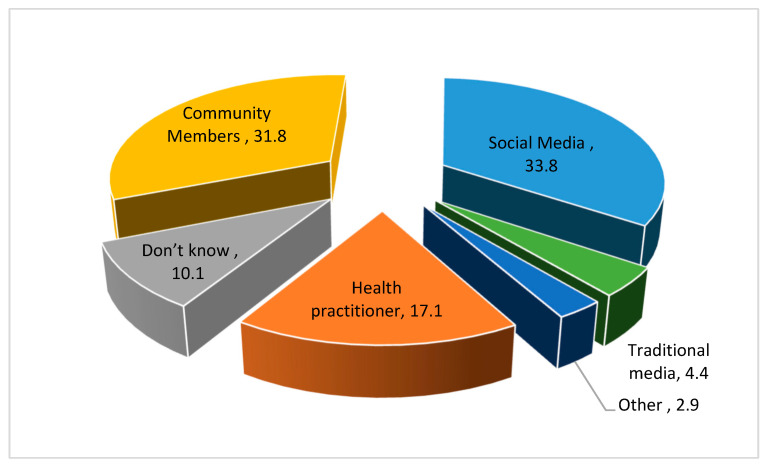
Source of information regarding tonsillectomy and autoimmune diseases.

**Figure 3 healthcare-11-00890-f003:**
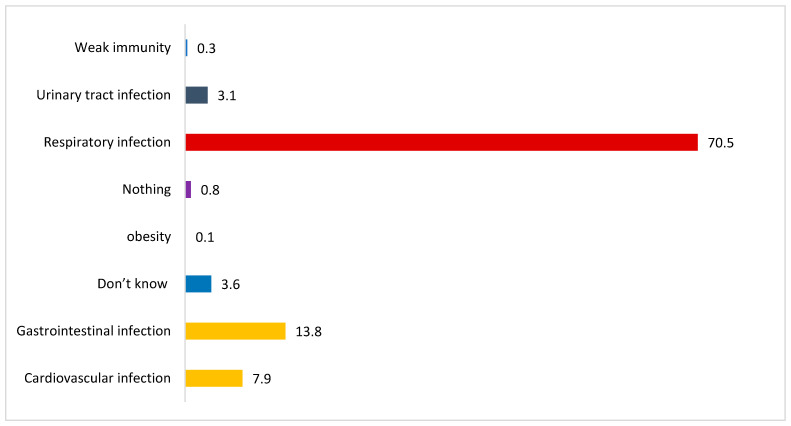
Perception regarding the type of infection that can affect a person after undergoing a tonsillectomy.

**Table 1 healthcare-11-00890-t001:** Association of sociodemographic factors and tonsillectomy.

Sociodemographic Factors	Category	Tonsillectomy		Total	*p* Value
		No (n = 696)	Yes (n = 104)		
Gender	Female	388	52	440	0.27
		88.2%	11.8%	100.0%	
	Male	308	52	360	
		85.6%	14.4%	100.0%	
Age group (in years)	Under 18	39	12	51	0.02
		76.5%	23.5%	100.0%	
	18 to 30	600	78	678	
		88.5%	11.5%	100.0%	
	31 to 40	38	8	46	
		82.6%	17.4%	100.0%	
	Older than 40	19	6	25	
		76.0%	24.0%	100.0%	
Education level	Middle school	8	11	19	0.00
		42.1%	57.9%	100.0%	
	High school	157	26	183	
		85.8%	14.2%	100.0%	
	Bachelors and above	531	67	598	
		88.8%	11.2%	100.0%	
Income (in SAR *)	Less than 5000	528	72	600	0.03
		88.0%	12.0%	100.0%	
	5000 to 9000	82	17	99	
		82.8%	17.2%	100.0%	
	More than 9000	86	15	101	
		85.1%	14.9%	100.0%	
Occupation	Unemployed	23	1	24	0.00
		96%	4%	100.0%	
	Employed	177	48	225	
		78.6%	21.4%	100.0%	
	Student	496	55	551	
		90.0%	10.0%	100.0%	

* SAR = Saudi Arabian Riyal.

**Table 2 healthcare-11-00890-t002:** Association of knowledge of respondents between tonsillectomy, immunity, and autoimmune diseases.

Query		Tonsillectomy		Total	*p* Value
		No (n = 696)	Yes (n = 104)		
Do you think tonsillectomy affects immunity?	I don’t know	220	20	240	0.01
		91.7%	8.3%	100.0%	
	No	228	47	275	
		82.9%	17.1%	100.0%	
	Yes	248	37	285	
		87.0%	13.0%	100.0%	
Do you think there is a relationship between tonsillectomy and autoimmune diseases (the body attacks itself)?	I don’t know	303	35	338	0.16
		89.6%	10.4%	100.0%	
	No	271	47	318	
		85.2%	14.8%	100.0%	
	Yes	122	22	144	
		84.7%	15.3%	100.0%	
Based on your knowledge of the relationship of tonsillectomy with immunity, have you decided not to get operated on yourself or any of your relatives?	No	562	84	646	0.99
		87.0%	13.0%	100.0%	
	Yes	134	20	154	
		87.0%	13.0%	100.0%	

**Table 3 healthcare-11-00890-t003:** Binary logistic regression analysis of demographic parameters contributing to tonsillectomy.

Sociodemographic Factors	Category	OR	95% CI		Tonsillectomy		Total	*p* Value
			Lower Limit	Upper Limit	Yes (n = 104)	No (n = 696)		
Gender	Female	Ref.			52	388	440	
	Male	0.79	0.52	1.19	52	308	360	0.27
Age group (in years)	Under 18	Ref.			12	39	51	
	18 to 30	2.36	1.18	4.71	78	600	678	0.01
	31 to 40	1.46	0.53	3.97	8	38	46	0.45
	Older than 40	0.97	0.31	2.99	6	19	25	0.96
Education level	Middle school	Ref.			11	8	19	
	High school	8.30	3.05	22.58	26	157	183	0.00
	Bachelors and above	10.89	4.23	28.05	67	531	598	0.00
Income	Less than 5000	Ref.			72	528	600	
	5000 to 9000	0.65	0.36	1.17	17	82	99	0.15
	More than 9000	0.78	0.42	1.42	15	86	101	0.42
Occupation	Unemployed and students	Ref.			56	519	575	
	Employed	0.39	0.26	0.60	48	177	225	0.00

OR: Odds ratio, CI: Confidence Interval.

## Data Availability

Not applicable.

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
