# Peer review of "Awareness and Perceptions of the Impact of Tonsillectomy on the Level of Immunity and Autoimmune Diseases among the Adult Population in Abha City, Kingdom of Saudi Arabia"

_healthcare, 2023, doi:10.3390/healthcare11060890_

Round 1

Reviewer 1 Report

It is a good work, but few specific comments are as follows:

Page 1, Line 35:  ENT………….Please add its full name before the abbreviation at the first time.

Page 1, Line 44:  tonsils…………………… change the font and remove the underline.

 Page 1-2, Lines 44-45:  Resentence to be clear………Recent literature also suggests that tonsils play an important role in the body’s immune system functioning.

References within the whole text must be written according to journal guidelines.

Page 2, Line 54:  Delete space.

Page 2, Line 62: It has been proven by performing certain laboratory tests……..Please mention those referred tests.

In the introduction section: The author repeats the same information many times, please delete the repetition.

Page 2, Lines 84-85: please check interfered words.   

In the introduction section: The author should mention the role or mechanism of tonsils in the immune system (in detail through B or T cells?) to know their side effect after surgery.  

Page 3, Line 115: The questionnaire contained the following items: 1- Demographic information, 2- Medical history of tonsillectomy, 3- Awareness Level………need more details about the questionnaire.

Ethical Considerations: Please mention the number of your ethical approval.

Table 1: Page 4

Bachelors and above

531

67

598

88.8

11.2

100.0

Add their percentage symbol.

Table 1, Page 4: Income by SAR or USD or what? Mention

 Page 5, Lines 157-159: Almost 36% of study subjects think that tonsillectomy affects immunity. Only 18 % of study subjects think that there is a relationship between tonsillectomy and autoimmune diseases………I didn’t find the referred percentage in Table 1, please check or it is collected data?

 Page 7, Line 200: Researchers suggested that the tonsils take part a great role in the evolution of…..resentence to be clear.

 Page 8, Line 243: Around 13% of our respondents who had tonsillectomy done before don’t want to suggest tonsillectomy for someone else based on its relationship to immunity………………..How do they know that tonsillectomy is related to immunity? Do they make any clinical examinations for that?

 cross-sectional online questionnaire survey was conducted among individuals 13 who were 18 years and above living in Abha city, Saudi Arabia……………..Author makes a questionnaire and uploads it as a google form online….that is right? All the participants lived in Abha city only. It is online and not face-to-face as the author mentioned.

 The author mentioned “ the relationship between autoimmune diseases and tonsillectomy”….I didn’t find any findings in the present study, what disease did u ask participants as an autoimmune disease?

 Only 3 questions for the present survey.

I thought the survey is too simple for judgment, according to the title.

Author Response

Reviewer 1

It is a good work, but few specific comments are as follows:

Page 1, Line 35:  ENT………….Please add its full name before the abbreviation at the first time.

Reply: As suggested the full name has been mentioned. (Please see line 36, 37)

Page 1, Line 44:  tonsils…………………… change the font and remove the underline.

Reply: As suggested this has been changed. (Please see line 46)

 Page 1-2, Lines 44-45:  Resentence to be clear………Recent literature also suggests that tonsils play an important role in the body’s immune system functioning.

Reply: As suggested this has been changed. (Please see line 46)

References within the whole text must be written according to journal guidelines.

Reply: As suggested this has been done.

Page 2, Line 54:  Delete space.

Reply: As suggested this has been done.

Page 2, Line 62: It has been proven by performing certain laboratory tests……..Please mention those referred tests.

Reply: As suggested this has been mentioned. (Please see line 64-65)

In the introduction section: The author repeats the same information many times, please delete the repetition.

Page 2, Lines 84-85: please check interfered words.  

Reply: As suggested this has been corrected. (Please see line 87-88)

In the introduction section: The author should mention the role or mechanism of tonsils in the immune system (in detail through B or T cells?) to know their side effect after surgery. 

Reply: As suggested this has been included. (Please see line 65-76)

Page 3, Line 115: The questionnaire contained the following items: 1- Demographic information, 2- Medical history of tonsillectomy, 3- Awareness Level………need more details about the questionnaire.

Reply: As suggested this has been mentioned. (Please see line 130-133)

Ethical Considerations: Please mention the number of your ethical approval.

Reply: As suggested this has been mentioned. (Please see line 141)

Table 1: Page 4

Bachelors and above

531

67

598

88.8

11.2

100.0

Add their percentage symbol.

Reply: As suggested this has been included. (Please see table 1)

Table 1, Page 4: Income by SAR or USD or what? Mention

Reply: As suggested this has been included. (Please see table 1)

 Page 5, Lines 157-159: Almost 36% of study subjects think that tonsillectomy affects immunity. Only 18 % of study subjects think that there is a relationship between tonsillectomy and autoimmune diseases………I didn’t find the referred percentage in Table 1, please check or it is collected data?

Reply: As pointed this has been corrected. (Please see table 2, lines 175-176)

 Page 7, Line 200: Researchers suggested that the tonsils take part a great role in the evolution of…..resentence to be clear.

Reply: As suggested this has been done. (Please line 218)

 Page 8, Line 243: Around 13% of our respondents who had tonsillectomy done before don’t want to suggest tonsillectomy for someone else based on its relationship to immunity………………..How do they know that tonsillectomy is related to immunity? Do they make any clinical examinations for that?

Reply: No comment

 cross-sectional online questionnaire survey was conducted among individuals 13 who were 18 years and above living in Abha city, Saudi Arabia……………..Author makes a questionnaire and uploads it as a google form online….that is right? All the participants lived in Abha city only. It is online and not face-to-face as the author mentioned.

Reply: Yes

 The author mentioned “ the relationship between autoimmune diseases and tonsillectomy”….I didn’t find any findings in the present study, what disease did u ask participants as an autoimmune disease?

Reply: Autoimmune diseases asked were IgA nephritis, IgA vasculitis, palmoplantar pustulosis, psoriasis, rheumatoid arthritis, Behçet’s disease, and myositis.

 Only 3 questions for the present survey.

I thought the survey is too simple for judgment, according to the title.’

Thank you for your valuable suggestions and helping us improve the manuscript.

Reviewer 2 Report

The authors present a set study based on a set of questionnaire on the awareness of tonsillectomy. The study shows that awareness is scarce and would require better information being disseminated. There are a few comments and question.

1. Income level with value should include the currency

2. Since the recruitment was done on-line, would it be a skewed/biased study?

3. Line 54 "going with tonsillectomy, 5357 of them were afterwards developed autoim-" there seem to be a missing word

4. Line 77-78 "Although available data is very scarce on public awareness, regarding effect of ton-77 sillectomy on immunity and development of autoimmune diseases; which influences the 78 decision of parents about this surgery." - sentence seemed to be hanging

5. Line 84-85 "about the relationship between tonsillectomy on the level of immunity and development of auto-84 immune diseases in the Kingdom of Saudi Arabia" there seems to be some sort of typo as the pdf generated showed no spacing

6. Sampling technique is simple random? or would it be better called as call for answer on a wide audience with subjects answering based on their own thoughts? Something more relevant/accurate?

Author Response

Reviewer 2

The authors present a set study based on a set of questionnaire on the awareness of tonsillectomy. The study shows that awareness is scarce and would require better information being disseminated. There are a few comments and question.

  1. Income level with value should include the currency

Reply: As suggested this has been included. (Please see table 1)

  1. Since the recruitment was done on-line, would it be a skewed/biased study?

Reply: As pointed by the respected reviewer we have mentioned this in the limitation. (Please see lines 277-278)

  1. Line 54 "going with tonsillectomy, 5357 of them were afterwards developed autoim-" there seem to be a missing word

Reply: As suggested this has been included. (lines 55-57)

  1. Line 77-78 "Although available data is very scarce on public awareness, regarding effect of ton-77 sillectomy on immunity and development of autoimmune diseases; which influences the 78 decision of parents about this surgery." - sentence seemed to be hanging

Reply: As suggested this has been corrected. (lines 91-93)

  1. Line 84-85 "about the relationship between tonsillectomy on the level of immunity and development of auto-84 immune diseases in the Kingdom of Saudi Arabia" there seems to be some sort of typo as the pdf generated showed no spacing

Reply: As suggested this has been corrected. (lines 97-100)

  1. Sampling technique is simple random? or would it be better called as call for answer on a wide audience with subjects answering based on their own thoughts? Something more relevant/accurate?

Reply: As suggested this has been corrected. (lines 135-136)

Thank you for your valuable suggestions and helping us improve the manuscript.

Reviewer 3 Report

First and foremost, the manuscript will benefit from extensive English language editing. In addition, the following issues have to be addressed:

1. Abstract, line 9 " leading to a deterioration of the situation". Please, specify which situation you mean.

2. Introduction, line 36 "In United States tonsillectomy is one of the commonest ENT surgical procedures as well." It is not clear why the authors from Saudi Arabia use the USA as a sample. They should refer to the neighboring countries instead.

3. Introduction, lines 40-47. This passages provides too much detail and can be omitted.

4. In general, Introduction can be shortened as many pieces of information may go to the Discussion.

5. Materials and methods, lines 94-97. The authors need to explain why they selected this particular city. 

6. Materials and methods. Did the authors calculate Cronbach's alpha?

7. Materials and methods, line 118. The authors have to explain the sampling strategy in detail. Earlier (line 110) the authors mentioned that the questionnaire was distributed via "social media and e-mail". It does not look like random sampling, but a convenience sampling. 

8. Ethical consideration. Please, indicate the date when permission was granted along with the protocol number.

9. Table 1. Indicate the currency for Income variable. Ideally, this should be transformed into one of the internationally recognized currencies.

10. How the authors will explain the fact that more than a half of their sample size was composed of students? Earlier, the authors stated that they utilized the simple random sample approach. Obviously, the authors cheat and this is a SCIENTIFIC MISCONDUCT.

11. Table 3. The authors have to indicate if they calculated crude or adjusted odds ratio. Please, address the issue of multicollinearity.

12. The figures are exhaustive, reduce the overall number to 2 figures.

13. Why some words in Discussion are underlined?

14. What are the study limitations? The authors need to consider them frankly. These limitations are obvious to the readers.

Author Response

The authors present a set study based on a set of questionnaire on the awareness of tonsillectomy. The study shows that awareness is scarce and would require better information being disseminated. There are a few comments and question.

  1. Income level with value should include the currency

Reply: As suggested this has been included. (Please see table 1)

  1. Since the recruitment was done on-line, would it be a skewed/biased study?

Reply: As pointed by the respected reviewer we have mentioned this in the limitation. (Please see lines 277-278)

  1. Line 54 "going with tonsillectomy, 5357 of them were afterwards developed autoim-" there seem to be a missing word

Reply: As suggested this has been included. (lines 55-57)

  1. Line 77-78 "Although available data is very scarce on public awareness, regarding effect of ton-77 sillectomy on immunity and development of autoimmune diseases; which influences the 78 decision of parents about this surgery." - sentence seemed to be hanging

Reply: As suggested this has been corrected. (lines 91-93)

  1. Line 84-85 "about the relationship between tonsillectomy on the level of immunity and development of auto-84 immune diseases in the Kingdom of Saudi Arabia" there seems to be some sort of typo as the pdf generated showed no spacing

Reply: As suggested this has been corrected. (lines 97-100)

  1. Sampling technique is simple random? or would it be better called as call for answer on a wide audience with subjects answering based on their own thoughts? Something more relevant/accurate?

Reply: As suggested this has been corrected. (lines 135-136)

Thank you for your valuable suggestions and helping us improve the manuscript.

Reviewer 4 Report

This study by Al-shaikh et al. conducted a survey regarding people’s awareness and perceptions of the relationship between tonsillectomy and immunity and autoimmune diseases after surgery in the adult population in Abha city. Overall, this study gives a new piece of information regarding an interesting area of study, however, several major points should be addressed to ensure the conclusions suggested by the authors are robust.

The following major points are raised:

1.     Have the authors conducted any survey regarding the reason(s) why people in Abha city pursued the tonsillectomy such as whether it is pathologic or non-pathologic etc., and how this reason(s) was raised such as whether a doctor or others recommend it? It is important as a starting point for this research.

2.     As the authors mentioned, the function of the tonsils is initiated between 3 to 10 years old of age, it is worth dissecting the under-18 group into more separated age groups, and the answers could be done by their lawful guardians, and the title should be revised accordingly.

Author Response

This study by Al-shaikh et al. conducted a survey regarding people’s awareness and perceptions of the relationship between tonsillectomy and immunity and autoimmune diseases after surgery in the adult population in Abha city. Overall, this study gives a new piece of information regarding an interesting area of study, however, several major points should be addressed to ensure the conclusions suggested by the authors are robust.

The following major points are raised:

  1. Have the authors conducted any survey regarding the reason(s) why people in Abha city pursued the tonsillectomy such as whether it is pathologic or non-pathologic etc., and how this reason(s) was raised such as whether a doctor or others recommend it? It is important as a starting point for this research.

Reply: Patients with Hypertrophy of tonsils and adenoids, chronic tonsillitis, adenoiditis and sleep disorders underwent tonsillectomy as recommend by a doctor. (Lines 133-135).

  1. As the authors mentioned, the function of the tonsils is initiated between 3 to 10 years old of age, it is worth dissecting the under-18 group into more separated age groups, and the answers could be done by their lawful guardians, and the title should be revised accordingly.

Reply: Unfortunately it is difficult for us to present this data. However as per the valuable suggestion we will try to attempt this in future studies.

Thank you for your valuable suggestions and helping us improve the manuscript.

Round 2

Reviewer 2 Report

Thank you for the corrections. 

Author Response

Thank you

Reviewer 3 Report

Dear authors, well done. 

My only suggestion is to add a footnote to Table 1 to explain abbreviation "SAR". Moreover, as the international readers may not be familiar with the ezchange rate, it wiould be helpful to include the corresponding amount in US dollars.

Author Response

My only suggestion is to add a footnote to Table 1 to explain abbreviation "SAR". Moreover, as the international readers may not be familiar with the ezchange rate, it wiould be helpful to include the corresponding amount in US dollars.

Reply: As suggested by the respected reviewer we have added the footnote. Thank you

Reviewer 4 Report

The authors have sufficiently addressed the comments raised in the last round of review.

Author Response

Thank you